# Quality and credibility of clinical practice guidelines recommendations for the management of neonatal hypoglycemia. A protocol for a systematic review and recommendations' synthesis

**Shaneela Shahid[1,2], Ginna Cabra-Bautista●[3], Ivan D. Florez●[4,5,6]***

**1** Department of Health Research Methods, Evidence, and Impact, McMaster University, Hamilton, ON, Canada, **2** Department of Pediatrics, McMaster University, Hamilton, Canada, **3** Department of Pediatrics, Universidad del Cauca, Popayán, Colombia, **4** Department of Pediatrics, University of Antioquia, Medellín, Colombia, **5** School of Rehabilitation Science, McMaster University, Hamilton, ON, Canada, **6** Pediatric Intensive Care Unit, Clinica Las Americas- AUNA, Medellin, Colombia

* ivan.florez@udea.edu.co

**Data Availability Statement:** No datasets were generated or analysed during the current study. All

## Abstract

### Introduction

Hypoglycemia is one of the most frequent metabolic conditions in neonates. Clinical practice guidelines (CPGs) influence clinical practice as high-quality CPGs facilitate the use of evidence in practice. This proposed study aims to systematically identify and appraise CPGs and CPG recommendations (CPGRs) for treating neonatal hypoglycemia (NH).

### Methods and analysis

We will conduct searches in MEDLINE, EMBASE, CINAHL, Cochrane Library, LILACS (Latin American & Caribbean Health Sciences Literature), and Epistemonikos. Authors will search CPGs-specific databases and grey literature. Two reviewers will independently perform the titles and abstract screening, full-text review, and data extraction. Two appraisers will assess the quality of the CPGs and their recommendations using AGREE II (Appraisal of Guidelines Research and Evaluation) and AGREE-REX (Appraisal of Guidelines Research and Evaluation–Recommendations Excellence) instruments. Scores of ≥ 60% in the rigour of development domain will be considered for defining high-quality with AGREE II tool. CPGRs with scores >60% in the three domains will be used to determine high quality with the AGREE REX tool. We will perform a synthesis of the CPGRs to identify the consistency among the CPGRs and the methodological quality of primary studies that support them.

### Ethics and dissemination

The results will help us to identify the methodological and quality gaps in the existing CPGs for the treatment of NH. Our findings will be submitted to peer-review journals and presented at academic conferences. Based on the study design, approval from the institutional ethics board is not required for this project.

relevant data from this study will be made available upon study completion.

**Funding:** The authors received no specific funding for this work.

**Competing interests:** The authors have declared that no competing interests exist.

## Trial registrations

Systematic Review Registration Number (PROSPERO): CRD 42021239921.

## Introduction

Glucose is an essential fuel for brain metabolism, and neonates have a high brain-to-body weight ratio [1] As such, they have a 2-to-3 fold higher glucose demand and utilization rate per kilogram of body weight than adults [2] Hypoglycemia is defined as low blood glucose concentration [3]. Blood glucose of less than 2.6 mmol/L has been considered an acceptable threshold for the treatment in this population [4, 5]. Neonatal hypoglycemia (NH) is one of the most frequently encountered problems in the first 48 hours of life [6], and its incidence has been estimated from 1 to 5 per 1,000 live births; however, the incidence in high-risk infants ranges from 5% to 15% [7]. Hypoglycemia in neonates may be associated with a risk for neurodevelopmental sequelae later [8–11]. Therefore, optimal management of these newborns with hypoglycemia is urgently required to prevent adverse neurological outcomes [12]. The most frequently used treatments for NH include breast or formula feeding, oral 40% dextrose gel, intravenous (IV) 10% or higher dextrose infusion, and glucagon [2, 6, 13, 14]. Since NH is a the widespread neonatal condition that requires immediate treatment, it is crucial to have clear, concise, and evidence-based clinical practice guidelines (CPGs) to help clinicians treat these nethe best way possible.

For CPGs to improve health outcomes, they must be developed using appropriate methods and rigorous development strategies and incorporate the highest quality of evidence [15–18]. Furthermore, scientific evidence of the benefits and harms of each available intervention is insufficient to ensure that the highest quality of the CPG recommendations have been achieved. Various other factors play an essential role in the guideline development process, including local practices of intended users, involvement of key stakeholders, availability of resources, strategies to improve the uptake of CPGs, patients' values and preferences, feasibility, and cost of the diagnostic tests or treatments [19, 20].

Several CPGs have been developed by different organizations and medical societies for the management of NH. However, these guidelines are not necessarily adopted by practicing physicians and hospitals as most of the centers or hospitals from the same region or country have their own CPG for the treatment of NH. This could be due to a lack of confidence in the evidence [21], conflicting CPGs [22, 23] or variation in the quality of CPGs. Therefore, the first step to understanding the guidelines is evaluating their quality and identifying the research gaps in the knowledge and in their development.

AGREE II (Appraisal of Guidelines Research and Evaluation) is the most widely used tool to assess the quality of CPGs. Moreover, this tool has been translated into several languages, and it has shown to be a valid and reliable tool [24, 25].The AGREE-REX (Appraisal of Guidelines Research and Evaluation–Recommendations EXcellence) is a tool recently developed by AGREE Collaboration that aims to assess the quality of the guideline's recommendations (credibility, trustworthiness, and applicability). In addition, this instrument is valid and reliable for assessing the quality of CPGs' recommendations [26]. Both tools are excellent resources for assessing CPGs and support researchers, users, and CPG developers in choosing the best guidelines or recommendations to be used or adapted in specific contexts.

The primary objective of this study is to systematically review, identify and appraise the quality of CPGs (position/consensus statements) and their recommendations for the treatment of NH that have been published in the last 20 years using both AGREE II and AGREE-REX

instruments. The secondary objectives of this review are to compare and contrast the recommendations of the various neonatal hypoglycemia CPGs and explore the differences among them regarding the evidence that supports the recommendations. We will also conduct a descriptive synthesis of the available clinical practice guideline recommendations (CPGRs) for the treatment of NH to describe the consistency among them and to map the methodological quality of primary studies that supported the recommendations.

## Methods

### Protocol and registration

The protocol for this SR has been submitted for registration to international databases of prospectively registered systematic reviews (PROSPERO) (registration number: CRD42021239921). This protocol was prepared in compliance with the Preferred Reporting Items for Systematic Review and Meta-Analysis Protocols (PRISMA-P) [27]. We will use the PICAR format (Population with clinical condition, Intervention, Comparator or no comparator, Attributes of eligible CPGs, and Recommendations characteristics) to formulate the research question, data collection, and data extraction [28]. The research question for this SR is: In neonates with hypoglycemia, do existing CPGs and their recommendations for the treatment of neonatal hypoglycemia have high methodological quality and credibility when assessed with the AGREE II and AGREE-REX instruments?

### Literature search

A systematic literature search will be conducted in the following databases: MEDLINE and EMBASE via Ovid; the Cochrane Library; CINAHL; LILACS, and Epistemonikos. Additional specific evidence-based or CPGs related websites or databases and specific scientific societies that will be searched include: TRIP (Turning Research into Practice), Health Systems Evidence, ECRI Guidelines Trust (Emergency Care Research Institute), WHO (World Health Organization); G-I-N (Guidelines International Network), NICE (National Institute for Health and Care Excellence), CMA (Canadian Medical Association), CADTH (Canadian Agency for Drugs & Technologies in Health), SIGN (Scottish Intercollegiate Guidelines Network), Australian CPGs, CENETEC (*Catálogo Maestro de Guías de Práctica Clínica*; México), *Guías de Práctica Clínica Ministerio de Salud y Protección Social* (Colombia), *Guías Clínicas* AUGE (Chile), *Guías Técnicas* (Perú), *Ministerio de Salud* (Argentina), Canadian Pediatric Society, and American Academy of Pediatrics.

We will also check the reference list of eligible CPGs and will use the snowballing technique to identify other possible CPGs. Medical Subject Headings in combination with keywords will be used for the following terms: neonates, newborn, blood glucose, hypoglycemia, recommendations, guidelines, consensus, and treatment. The search strategy in Medline is presented in the S1 File. The literature search will be limited to the last 20 years as the AGREE tool was first developed in 2004, and it is believed that the CPGs developed afterward, might have followed and adhered to the quality assessment criteria defined by the AGREE Collaboration.

### Eligibility criteria

We will include overall or specific CPGs, position or consensus statements on the treatment of NH, including pharmacological and non-pharmacological or oral and intravenous interventions. We will consider CPGs in English or Spanish, with two or more authors. The most up-to-date CPG from each organization/group will be included.

We will exclude documents that are incomplete or are the translation of CPGs in another language (aiming to include only the primary CPG) and anything that reports only an algorithm of management. Local hospitals' CPGs will also be excluded.

## Study selection

References will be managed in EndNote (version: 8.2), and duplicates will be removed. Published guidelines and guidelines from complementary searches will be included if they treat neonates with hypoglycemia. We will use the PICAR format (Population, Intervention, Comparator/Comparison/Content, Attributes of eligible CPGs, and Recommendation characteristics) to define CPGs and CPGRs eligibility criteria [28]. Title and abstracts will be independently screened by two reviewers. Citations deemed to be relevant will be passed on to the next stage. The same reviewers will conduct a full review of the CPGs independently to confirm the eligibility of the guidelines. The most up-to-date guidelines will be used if there is a case of duplicate publications. Any disagreements which occur will be resolved through discussion/consensus. The Preferred Reporting Items for Systematic Reviews (PRISMA) flow diagram will be used to show the studies selection process [29, 30].

## Data extraction

The data extraction process will be conducted by two reviewers independently and will be extracted on a Microsoft Excel spreadsheet. The PICAR format [28] will guide the data collection and data extraction. We will extract the following information from each CPG: population of interest, type of hypoglycemia, year of publication, author team, name of the organization, scope of CPG, the purpose of CPG, type of CPG, the country of development, intended users, search databases, methods of recommendation formulation, methods for classifying the quality of evidence, recommendation characteristics, treatment's types, and specific recommendations made by the CPG on each treatment of NH. Data will also be collected on evidence of benefits and harms, feasibility, values/preferences, and cost of recommended treatments. With CPGs that reported treatment of hypoglycemia for various ages, only recommendations regarding neonates will be considered.

## Quality appraisal of CPGs and recommendations

Two approaches are described by the AGREE research group to assess the quality of CPGs using AGREE II and AGREE-REX instruments [31]. The first approach is suggested to be used when researchers have resources and time and when high-stake health care decisions need to be made. This approach includes assessing the quality of all the eligible CPGs using all domains of AGREE II and then moving to AGREE-REX. The second approach is the fast-track approach, which has five different strategies and is considered when time and resources are limited. Since there are no publications on the quality of the assessment of CPGs for NH, we have decided to implement the first approach. Using the AGREE-REX tool, we will assess the quality of all CPGRs made by included CPGs instead of one or specific CPGRs.

The AGREE instrument was developed in 2003 [32] and updated as AGREE II in 2010 [33]. It measures the methodological rigor and transparency of the development of the CPGs [34] AGREE II includes six domains: scope and purpose, stakeholder involvement, rigor of development, clarity of presentation, applicability, and editorial independence, and contains 23 items [33] in total. The quality score for each item is on the Likert scale of 1 to 7 (totally disagree to totally agree). A final item requires the assessment of the overall CPG quality and determining whether the CPG can be recommended for use in practice [34]. AGREE II instrument is a valid and reliable tool that has been widely used to assess the quality of CPGs.

However, it has been found that having high-quality CPGs is essential but not sufficient to ensure adequate quality of the recommendations. As such, the AGREE-REX instrument was recently developed to evaluate the quality of CPGRs specifically, defined as credible and implementable recommendations [35]. AGREE-REX instrument is also a valid and reliable tool, and it has three domains (applicability, values/preferences, and implementability) containing nine items [26, 36, 37]. Each item has a list of criteria to assess the item's quality on the Likert scale of 1 to 7 (lowest quality to highest quality). The tool has two final questions for the appraisers to decide whether they would recommend the guideline or not in the appropriate context and for the context in which it is aimed to be applied by the appraisers [26, 36, 37].

The AGREE Collaboration recommends at least two but preferably four appraisers to assess the quality of CPGs and their recommendations to improve reliability [24, 34]. For this review, two appraisers will independently perform a quality assessment using AGREE II and AGREE-REX instruments independently. Upon completion of the appraisal, scores from each domain, as well as an overall evaluation, will be reviewed by the corresponding author, and any discrepancies in the score of 3 points or greater will be flagged and discussed, and a consensus on the appropriate score will be reached.

## AGREE II and AGREE-REX training

The appraisers will be trained to use the AGREE II and AGREE-REX instruments in conjunction with the AGREE II and AGREE-REX manuals; register with My AGREE PLUS platform, which is available at http://www.agreetrust.org/resource-centre/agree-plus/; and fill out the AGREE II Online Training Tool available at http://www.agreetrust.org/resource-centre/agree-ii-training-tools/. Appraisers will also evaluate two CPGs of different levels of quality for the treatment of NH using AGREE II and AGREE-REX instruments. They will discuss the results with the senior trained appraiser.

## Data synthesis and analysis

All statistical analyses will be conducted using SPSS (IBM SPSS Statistics Version 26.0, NY). Agreement between reviewers for the title and abstract screening and full-text review for CPG eligibility will be calculated using Cohen's kappa statistic. Agreement between the appraisers for AGREE II and AGREE- REX tools will be calculated using the intra-class coefficient (ICC) with a two-way mixed model and stratified as poor ($<0.20$); fair ($0.21–0.41$); moderate ($0.41–0.60$); good ($0.61–0.80$); and very good ($0.81–1.00$) [38]. A statistical analysis will be conducted by calculating scores for each AGREE II and AGREE-REX domain. Following the AGREE II and AGREE-REX manual instructions [33, 36], domain scores for both tools will be calculated by adding the scores of individual items within a domain, then scaling the total as a percentage of the maximum possible score for that domain. We will calculate the scores as follows: (obtained score—minimal possible score)/(maximal possible score—minimal possible score) multiplied by 100 to get the scaled percentage of the maximum possible score. A score of 0% represents the worst and 100% the best possible rating for each domain, respectively.

We will calculate the overall scores of the included CPGs for each AGREE II and AGREE-REX domain using summary statistics, i.e., mean and the standard deviation (SD), and median and interquartile range (IQR). Results of domain quality scores will be reported in the median (IQR). The AGREE collaboration does not set a threshold of the AGREE II tool for categorizing a guideline as of high quality and advises users to determine their own quality threshold according to the context and preferences [31]. Although several approaches have been used, a score of $\geq$60% in the third (Rigour of development) domain has been commonly used to define CPGs as of high quality [39]. This threshold has been used in systematic reviews

of CPGs on other health-related topics in neonates [40–42], and as such, we decided to implement it in this project. Moreover, we will also use the two final assessment questions of the AGREE II tool, namely, the overall quality of CPGs and whether CPG can be recommended in practice, to define whether we would recommend the CPG to be used in clinical practice. Since no specific guidance or threshold has been established for this final assessment, we will use a modified approach of the criteria used by Cabra-Bautista et al. [43] but using the rigor of development as the primary domain to consider. Therefore, a CPG will be 'recommended' if four (rigor of development being one of the domains) out of six domains score ≥60%; a CPG will be 'recommended with modifications' if it does not meet the criteria for being 'recommended' or 'not recommended'; and will be considered 'not recommended' when three out of six domain scores <30% or the domain three scores <60%.

AGREE-REX is a new tool; therefore, minimal guidance is available from the literature about the quality threshold score for CPGRs. As such, we cannot prioritize a specific domain over the others. We will, therefore, use the domain score of ≥ 60% in all three domains as the threshold for defining high-quality CPGRs [26, 36, 37]. For the final question as to whether CPGRs can be recommended or not, we will define it as 'recommended' if they score ≥60% in the three domains; as 'recommended with modifications' if it does not meet the criteria for being 'recommended' or 'not recommended'; and, as 'not recommended' if the two or more domains score <30%.

## Synthesis of clinical practice guideline recommendations on interventions (CPGRs)

We will conduct the descriptive synthesis to describe the consistency in CPGRs for the treatment of NH. Furthermore, we will map the methodological quality of the primary studies that supported the recommendations. We will create an evidence matrix based on the risk of bias (RoB) and the quality of studies as suggested by Johnston et al. [28]. We will use the following tools and resources to assess the quality of the primary studies supporting the recommendations: Cochrane RoB (Risk of Bias) 2.0 for Randomized Controlled Trials (RCTs) [44], ROBIS (A Risk of Bias Assessment Tools for Systematic Reviews) for SR [45], and ROBINS-I (Risk of Bias In Non-randomised Studies of Interventions-NRSI) for non-randomized studies focused on interventions [46].

Although the evidence for supporting recommendations on interventions should come from RCTs, NRSI, or SR of them, we acknowledge that it may not be the case in some guidelines. Thus, we are interested in assessing the quality of their information sources. When authors support their recommendations on case series or cross-sectional studies, we will assess their quality with the JBI (Joanna Briggs Institute) tool [47, 48], and if narrative reviews are used, we will assess them with SANRA (Scale for the quality assessment of narrative reviews) [49].

Since this study focuses on interventions for NH, it is important to highlight the level of evidence used to formulate the recommendation for each intervention which will help the clinicians and end-users of the guidelines to compare the intervention based on the level of evidence when treating infants with neonatal hypoglycemia. As such, we plan to create an evidence matrix using the highest level of evidence (SR, RCT, and non-randomized studies) for each intervention recommended by the CPGs for the treatment of NH.

## Discussion

This review will evaluate the quality of CPGs using AGREE II and their recommendations using AGREE-REX instruments. We will also identify and evaluate the evidence underpinning

their recommendations. This review will highlight the methodological standards used during the CPGs development and CPGRs formulation and the transparency of reporting of these CPGs for the treatment of NH. Moreover, it will inform future researchers about the quality of primary studies used in the formulation of recommendations and how the quality of the evidence for the treatment of NH can be improved by conducting high-quality studies comparing different interventions. We believe this review will highlight and identify high quality CPGs and CPGRs on NT, which will greatly interest CPG developers, policymakers, health care organizations and medical societies worldwide. Moreoverif the organization or medical society decide to adopt/adapt from the existing CPGs and CPGRs on NT, this review will provide them guidance about high quality CPGs and CPGRs on NT. Research findings from this review will identify the methodological and quality gaps in existing guidelines which will help methodologists and CPGs developers of future CPGs. This review will inform clinicians on which high-quality CPGs and CPGRs could be used in their clinical practice for the treatment of NH.

## Supporting information

**S1 Checklist. PRISMA-P 2015 checklist.**
(PDF)

**S1 File.**
(DOCX)

## Acknowledgments

The authors would like to thank Peter Rosenbaum for his helpful comments on the proposal of this study.

## Author Contributions

**Conceptualization:** Shaneela Shahid, Ivan D. Florez.

**Data curation:** Shaneela Shahid.

**Formal analysis:** Shaneela Shahid, Ivan D. Florez.

**Investigation:** Shaneela Shahid, Ginna Cabra-Bautista, Ivan D. Florez.

**Methodology:** Shaneela Shahid, Ginna Cabra-Bautista, Ivan D. Florez.

**Project administration:** Shaneela Shahid.

**Supervision:** Ivan D. Florez.

**Validation:** Shaneela Shahid, Ginna Cabra-Bautista.

**Writing – original draft:** Shaneela Shahid, Ivan D. Florez.

**Writing – review & editing:** Shaneela Shahid, Ginna Cabra-Bautista, Ivan D. Florez.

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
