## [Decision Letter · Decision Letter 0]

1 Sep 2022

PONE-D-22-19339Quality and Credibility of Clinical Practice Guidelines Recommendations for the Management of Neonatal Hypoglycemia. A Protocol for a Systematic Review and Meta-synthesisPLOS ONE

Dear Dr. Florez,

Thank you for submitting your manuscript to PLOS ONE. After careful consideration, we feel that it has merit but does not fully meet PLOS ONE’s publication criteria as it currently stands. Therefore, we invite you to submit a revised version of the manuscript that addresses the points raised during the review process.

Minor revisions requested. Please respond clearly to all peer-review and editor comments. 

We look forward to receiving your revised manuscript.

Kind regards,

Dr Michael McCaul, MSc, PhD

Academic Editor

PLOS ONE

Journal Requirements:

Additional Editor Comments:

The authors propose a SR and meta-synthesis of available CPGs, where synthesis occurs at recommendation level.

From the description of the meta-synthesis the 'unit of analysis' in the meta-synthesis is not clear to me. Neither is it clear what meta-synthesis methods are being used or where such methods has been used before.

I encourage the authors to clarify this important distinction (including references), which will set this reviews methods apart from a scoping review. The secondary objective is still descriptive in nature and not clearly linked to the synthesis or pooling concept.

Reviewers' comments:

Reviewer's Responses to Questions

**Comments to the Author**

1. Does the manuscript provide a valid rationale for the proposed study, with clearly identified and justified research questions?

Reviewer #1: Yes

Reviewer #2: Yes

2. Is the protocol technically sound and planned in a manner that will lead to a meaningful outcome and allow testing the stated hypotheses?

Reviewer #1: Yes

Reviewer #2: Yes

3. Is the methodology feasible and described in sufficient detail to allow the work to be replicable?

Reviewer #1: Yes

Reviewer #2: Yes

4. Have the authors described where all data underlying the findings will be made available when the study is complete?

Reviewer #1: No

Reviewer #2: Yes

5. Is the manuscript presented in an intelligible fashion and written in standard English?

Reviewer #1: No

Reviewer #2: Yes

6. Review Comments to the Author

You may also provide optional suggestions and comments to authors that they might find helpful in planning their study.

Reviewer #1: Thank you for this protocol - it was interesting to read and the results will be valuable to those managing neonatal hypoglycaemia, as well as an approach to assessing CPGs of different health conditions and identifying gaps. Some feedback:

Q4: data sharing stated as non applicable as this is a protocol. It would be helpful to know where the data will be found when the study is complete.

Q5: In general the writing is really good. Just a few suggestions: line 228 - sentence starting with "Furthermore..." is not clear and should be rewritten.

Line 234 - the words in brackets should be reordered to make sense e.g. (domain three being one of them)

Line 236 - should be "domains score" instead "when three out of six domain scores"

Line 250-1 - assume this is Cochrane RoB 2.0 tool? In which case correct here

Other comments on clarification: in the discussion, how will this research assist with adoption/adaption of high-quality CPGs (line 276-7) - would organizations not just need to do AGREE II and AGREE-REX of their own GL? I think this sentence perhaps needs clarification as it is not really clear how this will be done, also to which high-quality CPGs/CPGRs you are referring to here - do you plan to share the outcome of each AGREE II and -REX with the organizations who published those guidelines?

Lines 278 to 280 are clear and make sense.

References: references are meant to be in Vancouver format (https://journals.plos.org/plosone/s/submission-guidelines) - there are a number of issues with your references:

* Reference should have a doi or web link ("Available from"), where possible

* Correct journal abbreviations should be used (hardly used any journal abbreviations, except Cmaj which should actually be Can. Med. Assoc. J.)

* Some references are incomplete (e.g. #17)

* Vancouver format has not be used in a standardised way e.g. Ref #24 lists 10 authors whilst Vancouver instructs to record the first 6 authors followed by "et al."

I suggest a full relook at all the references to ensure they are in the correct format. Guidelines are available in the "instructions to authors"

Reviewer #2: The protocol is well described and is well consistent with the purpose of the study.

I have only three points for the authors to consider:

a) wouldn't the search be more appropriate in the Cochrane Library rather than in CENTRAL since the latter is specific for clinical trials?

b) in the abstract there is no reference to LILACS and Epistemonikos - will the search be performed in these databases?

c) in the introduction section (lines 89 to 94), it seems that the two "goals" are repeated and neither of them refers to the research question indicated in the methods section.

7. PLOS authors have the option to publish the peer review history of their article (what does this mean?). If published, this will include your full peer review and any attached files.

Reviewer #1: No

Reviewer #2: **Yes: **DANIELA OLIVEIRA DE MELO

---

## [Author Response · Author response to Decision Letter 0]

17 Nov 2022

Response to the Editor

The authors propose a SR and meta-synthesis of available CPGs, where synthesis occurs at recommendation level.From the description of the meta-synthesis the 'unit of analysis' in the meta-synthesis is not clear to me. Neither is it clear what meta-synthesis methods are being used or where such methods has been used before.

I encourage the authors to clarify this important distinction (including references), which will set this reviews methods apart from a scoping review. The secondary objective is still descriptive in nature and not clearly linked to the synthesis or pooling concept. 

Response: Thanks for your comment. You are right; according to our proposal, our approach should not be considered a meta-synthesis, as this method has been mostly described to synthesize evidence from qualitative literature. We aimed to synthesize the available recommendations describing them using tables and evaluating the evidence and the disussions underpinning them. We have made this change in the manuscript and have removed the meta-asynthesis term, stating this is a descriptive synthesis of the recommendations

Response to Reviewers

1. Does the manuscript provide a valid rationale for the proposed study, with clearly identified and justified research questions?

Response: Thank you for the “Yes” comment by both reviewers.

2. Is the protocol technically sound and planned in a manner that will lead to a meaningful outcome and allow testing of the stated hypotheses?

Response: We appreciate that both reviewers commented “Yes” to this question.

3. Is the methodology feasible and described in sufficient detail to allow the work to be replicable?

Response: Thank you for the “Yes” comment by both reviewers.

4. Have the authors described where all data underlying the findings will be made available when the study is complete?

 Response: We appreciate your feedback. We have modified the Data availability statement on page 14, lines 292 – 293.

5. Is the manuscript presented in an intelligible fashion and written in standard English?

Response: We appreciate your constructive feedback and this manuscript was proofread again to meet the Journal requirement.

Reviewer 1: 

Line 228 - sentence starting with "Furthermore..." is not clear and should be rewritten 

Response; Thank you for this feedback. This statement is now on line 232 and is modified to “Moreover, we will also use the two final assessment questions of the AGREE II tool, namely, the overall quality of CPGs and whether CPG can be recommended in practice, to define whether we would recommend the CPG to be used in clinical practice”.

Line 234 - the words in brackets should be reordered to make sense e.g. (domain three being one of them)

Response: Your constructive feedback was highly appreciated. This statement is now on line 238 and is modified to “Therefore, a CPG will be 'recommended' if four (rigor of development being one of the domains) out of six domains score ≥60% 

Line 236 - should be "domains score" instead "when three out of six domain scores"

Response: Thanks for your comment. There is not one single rule on to how to define high quality. As a result, many authors have used differnet thresholds for the domains and with a specific threshold, many approaches have been implemented. We thought that less than 30% in all the domains is a very extreme situation, and having <30% in one domain and may occur in domains that are very contextual (such as applicability domain). Therfore, we chose a point in the middle: havind half of somains below 30% will be a good threshold of a guideline uf very low quality and thus, should not be recommended. Thus, no changes are applied in this point

Line 250-1 - assume this is Cochrane RoB 2.0 tool? In which case correct here

Response: Thank you for this comment, this statement is now on line 256, and it is correct that Cochrane RoB 2.0 tool will be used. 

In the discussion, how will this research assist with the adoption/adaption of high-quality CPGs (line 276-7) - would organizations not just need to do AGREE II and AGREE-REX of their own GL? I think this sentence perhaps needs clarification as it is not really clear how this will be done, also to which high-quality CPGs/CPGRs you are referring here - do you plan to share the outcome of each AGREE II and AGREE-REX with the organizations who published those guidelines?

Response: Thank you for this comment. This review will identify which CPGs and CPGRs on NT are high-quality based on AGREE II and AGREE-REX tools. The development of CPGs requires resources and expertise which might not be possible for low-income or middle-income countries as such having identification of high-quality CPGs and CPGRs will provide guidance to the organizations/medical societies of these countries. 

Based on your feedback, we have modified it to “We believe this review will highlight and identify high-quality CPGs and CPGRs on NT, which will greatly interest CPG developers, policymakers, health care organizations, and medical societies worldwide. Moreover, if the organization or medical society decides to adopt/adapt the CPGs and CPGRs on NT, this review will provide them guidance about high-quality CPGs and CPGRs on NT.” This statement is now from line 280 to line 284.

 Lines 278 to 280 are clear and make sense.

 Response: We appreciate your positive feedback. 

Reviewer 2: 

Wouldn't the search be more appropriate in the Cochrane Library rather than in CENTRAL since the latter is specific for clinical trials?

Response: Thank for this feedback, we have modified from CENTRAL to Cochrane Library, page 2, line 32-33 and page 6, line 115-116. We will do the search using the Cochrane Library search engine instead of CENTRAL. 

In the abstract there is no reference to LILACS and Epistemonikos - will the search be performed in these databases?

Response: We appreciate this comment, we will use LILACS and Epistemonikos databases will be used and we have included them in the abstract, page 2, line 32 to 33. 

In the introduction section (lines 89 to 94), it seems that the two "goals" are repeated and neither of them refers to the research question indicated in the methods section.

Response: We appreciate your constructive feedback. Based on your feedback, we have modified the statement about the primary objective of this review. Our first goal is to systematically review, identify and appraise the quality of CPGs (position/consensus statements) and their recommendations for the treatment of NH using both AGREE II and AGREE-REX instruments, refers to the methodological quality and credibility of the CPGs/CPGRs. We have made changes to our second goal as a secondary objective. Now we have one primary objective and a couple of secondary objectives. Please refer to pages 5-6 and lines 90 to 100.

---

## [Decision Letter · Decision Letter 1]

4 Jan 2023

Quality and Credibility of Clinical Practice Guidelines Recommendations for the Management of Neonatal Hypoglycemia. A Protocol for a Systematic Review and Recommendations' Synthesis

PONE-D-22-19339R1

Dear Dr. Ivan 

We’re pleased to inform you that your manuscript has been judged scientifically suitable for publication and will be formally accepted for publication once it meets all outstanding technical requirements.

Kind regards,

Edris Hasanpoor

Academic Editor

PLOS ONE

---

## [Editor Report · Acceptance letter]

10 Jan 2023

PONE-D-22-19339R1 

Quality and Credibility of Clinical Practice Guidelines Recommendations for the Management of Neonatal Hypoglycemia. A Protocol for a Systematic Review and Recommendations’ synthesis 

Dear Dr. Florez:

I'm pleased to inform you that your manuscript has been deemed suitable for publication in PLOS ONE. Congratulations! Your manuscript is now with our production department. 

Kind regards, 

on behalf of

Dr. Edris Hasanpoor 

Academic Editor

PLOS ONE